# Fingolimod inhibits multiple stages of the HIV-1 life cycle

**Rachel S. Resop**[1], **Rémi Fromentin**[2], **Daniel Newman**[1], **Hawley Rigsby**[2], **Larisa Dubrovsky**[1], **Michael Bukrinsky**[1], **Nicolas Chomont**[2], **Alberto Bosque**[1]*

**1** Department of Microbiology, Immunology and Tropical Medicine, The George Washington University, Washington, D.C., United States of America, **2** Centre de recherche du CHUM and Department of microbiology, infectiology and immunology, Université de Montréal, Montreal, Canada

* abosque@email.gwu.edu

**Data Availability Statement:** All relevant data are within the manuscript and its Supporting Information files.

**Funding:** Research reported in this publication was partially supported by the NIAID (https://www.

## Abstract

Antiretroviral drugs that target various stages of the Human Immunodeficiency Virus (HIV) life cycle have been effective in curbing the AIDS epidemic. However, drug resistance, off-target effects of antiretroviral therapy (ART), and varying efficacy in prevention underscore the need to develop novel and alternative therapeutics. In this study, we investigated whether targeting the signaling molecule Sphingosine-1-phosphate (S1P) would inhibit HIV-1 infection and generation of the latent reservoir in primary CD4 T cells. We show that FTY720 (Fingolimod), an FDA-approved functional antagonist of S1P receptors, blocks cell-free and cell-to-cell transmission of HIV and consequently reduces detectable latent virus. Mechanistically, FTY720 impacts the HIV-1 life cycle at two levels. Firstly, FTY720 reduces the surface density of CD4, thereby inhibiting viral binding and fusion. Secondly, FTY720 decreases the phosphorylation of the innate HIV restriction factor SAMHD1 which is associated with reduced levels of total and integrated HIV, while reducing the expression of Cyclin D3. In conclusion, targeting the S1P pathway with FTY720 could be a novel strategy to inhibit HIV replication and reduce the seeding of the latent reservoir.

## Author summary

Human Immunodeficiency Virus (HIV) is currently managed by antiretroviral drugs, which may have side effects and are of limited use in prevention of transmission of the virus between individuals. We investigated an alternative tactic to combat HIV infection by harnessing a component of the immune system involved in the progression of infection, Sphingosine-1-phosphate (S1P). We tested a drug known to modulate the action of S1P receptors, FTY720 (Fingolimod) in human immune cells to investigate whether targeting S1P could inhibit HIV infection. We observed that FTY720 was able to block infection in human CD4 T cells by hindering multiple steps in the life cycle of HIV. FTY720 is already clinically approved and did not affect the viability of the human cells in our model system; therefore, we believe that this compound may be a promising novel therapy for HIV treatment and prevention.

niaid.nih.gov/) of the NIH under grants R01-AI124722, R21/R33-AI116212 to A.B. This research has been facilitated by the services and resources provided by District of Columbia, Center for AIDS research, an NIH-funded program (AI117970), which is supported by the following NIH cofunding and participating institutes and centers: NIAID, NCI, NICJD, NHLBI, NIDA, NINH, NIA, FIC, NIGGIS, NDDK, and OAR (https://dccfar.gwu.edu/). The content is solely the responsibility of the authors and does not necessarily represent the official views of the NIH. The funders had no role in study design, data collection and analysis, decision to publish, or preparation of the manuscript.

**Competing interests:** The authors have declared that no competing interests exist.

## Introduction

Human Immunodeficiency Virus (HIV-1) remains a global health burden, with nearly 40 million people currently living with HIV. Treatment of infection is lifelong, due to the ability of the virus to establish latency by integrating its genome into that of host cells, which become quiescent and long-lived, resulting in the potential of viral reactivation at a future time [1–8]. By establishing latency, HIV evades eradication by host defense mechanisms and drug treatment. Following various stimuli including immune activation and "kick and kill" strategies, quiescent cells harboring virus become activated and the virus becomes transcriptionally active, resulting in production of viral proteins [9, 10]. Currently available antiretroviral therapies (ART) are able to control viral load but do not specifically target latent infection and have off-target effects in many individuals [11]. For this reason, discovery of novel methods to target establishment of HIV infection and latency is crucial.

Currently implemented ART consists mainly of classes of drugs that target various stages of the viral life cycle, including inhibitors of entry, protease, integrase and reverse transcriptase [11]. Another potential target to treat HIV infection is the use of immunomodulatory compounds directed toward a component of the immune system. This tactic could potentially have efficacy in a wide range of individuals and avoid some of the off-target effects observed with ART. Thus far, immunomodulatory compounds that target innate immune factors have not been extensively characterized for treatment of HIV. We were interested in examining the effect of modulation of the cellular signaling molecule Sphingosine-1-phosphate (S1P) on the establishment of productive as well as latent HIV infection. S1P is a lysophospholipid intra- and intercellular signaling molecule with a myriad of roles in the human body, including cell proliferation and migration, cytoskeleton rearrangement, membrane integrity, adhesion, survival/ apoptosis, and inflammation in nearly all cell types. These processes are modulated through the five receptors to S1P, known as S1PR1-5 (reviewed in [12]). Although it is known that S1P receptor 1 (S1PR1) and S1P receptor 4 (S1PR4) are expressed on several subsets of CD4 T cells and can be modulated by various agonists and specific antagonists [13–15], the full spectrum of functions of S1P signaling in CD4 T cells, as well as the impact of HIV-1 infection on expression and activity of the S1P receptors, remains to be elucidated. The involvement of S1P in inflammation and various diseases is established [12, 16, 17] and S1P signaling modulators have been studied extensively as potential cancer treatments [18–20], yet there is a paucity of knowledge of the role of S1P in HIV-1 pathogenesis.

FTY720, also known as Fingolimod or Gilenya, is an immunomodulatory compound that acts as a Sphingosine-1-phosphate receptor (S1PR) non-selective agonist and a selective antagonist of S1PR1 [21]. FTY720 was initially synthesized using the naturally occurring fungal compound myriocin (ISP-1) as a lead [22]. The compound is clinically approved for treatment of Multiple Sclerosis [23, 24] and is well-tolerated when taken orally on a daily basis [25, 26]. FTY720 has activity at four of the five S1P receptors (S1PR1, 3, 4 and 5) and has been shown to cause downregulation of S1PR1 in lymphocytes and act as a modulator of S1P signaling, effecting changes in chemotaxis, proliferation, and cell cycle state of lymphocytes and other cells [15, 21, 27–29]. In the case of S1PR1, FTY720 binds to this receptor when phosphorylated and causes its internalization and loss of signaling [27].

It has been reported that S1PR1 is highly co-expressed with the HIV-1 coreceptor CCR5 on CD4 T cells and that targeting of S1PR1 reactivates HIV-1 from latency in an NF-κB-dependent manner [14]. Recently, FTY720 has been shown to promote the retention of T cells in the lymph node in SIV-infected non-human primates with a subsequent reduction in the levels of SIV DNA in the blood [30]. Further indicating an interplay between S1P signaling and progression of HIV infection, inhibition of glycosphingolipid metabolism was shown to impact

susceptibility of CD4 T cells to infection [31], and an impaired response to S1P and altered Akt signaling of lymph node CD4 T cells in chronically infected HIV-1 patients has also been observed [32]. Due to the established clinical efficacy and safety of FTY720, we hypothesized that this immunomodulatory compound could potentially inhibit HIV infection. In this work, we report that modulation of the S1P receptors with FTY720 reduces the susceptibility of CD4 T cells to both productive and latent HIV infection in primary CD4 T cells and suggest the use of FTY720 as a potential novel adjuvant to treat HIV infection in prevention as well as curative strategies.

## Results

### Functional antagonism of S1P signaling inhibits cell-free infection of HIV-1

We first investigated the involvement of the S1P signaling pathway on establishment of productive HIV infection using a modification of our primary cell model of HIV (**Fig 1A**), [33–35]. We isolated naïve CD4 T cells from PBMCs of HIV-negative human donors by negative selection and activated with αCD3/28 as previously described [33]. Following activation, cells were expanded in culture with IL-2. To address the effects of FTY720 on cell-free HIV infection (**Fig 1A**, left schematic), CD4 T cells were treated with FTY720 at day 5 of culture at a range of concentrations (30-100nM) for 48 hours. At day 7, cells were infected with either an X4- or R5-tropic HIV-1 (NL4-3 or NL-AD8, respectively). We assessed levels of infection by measuring p24-gag by flow cytometry 72 hours later (day 10). A representative flow cytometry analysis (of 7 donors, NL4-3 infected) is presented in **Fig 1B**. We found that pre-treatment with FTY720 reduced HIV-1 infection in a dose-dependent manner with an average reduction of the frequency of infected cells for NL4-3 of 28.70+/-16.69%, 39.53+/-9.42%, 45.34+/-2.85% and 55.23+/-7.57% for 30nM, 44nM, 66nM and 100nM FTY720, respectively (**Fig 1C**), and an average reduction of the frequency of infected cells for NL-AD8 of 31.75+/-13.91%, 32.16 +/-9.95%, 41.60+/-12.57% and 41.12+/- 8.21% for 30nM, 44nM, 66nM and 100nM FTY720, respectively (**Fig 1D**). Treatment with FTY720 did not alter viability at any of the concentrations examined (**S1A and S1B Fig**) and the inhibitory efficacy of FTY720 was independent of viral tropism or strain (dose response curve, **Fig 1E**) and **S2 Fig**.

Next, we investigated the effect of FTY720 on cell-to-cell transmission. To simulate cell-to-cell transmission, HIV-infected primary CD4 T cells were cultured in 96-well round bottom plates for three days (days 10–13, **Fig 1A**, right schematic). We have previously shown that this procedure enhances cell-to-cell HIV transmission in primary CD4 T cells [34]. In line with our cell-free infection results, we observed a marked decrease in productive infection with 66nM FTY720 following three days of culture as assessed by p24 at day 13 (**Fig 1F**), with an average reduction in p24 of 54.4+/-16.02% (**Fig 1G and 1H**, raw values and relative infection, respectively). This inhibition was, as for cell-free infection, not restricted to X4-tropic virus, as we saw a similar inhibition of cell-to-cell transmission with R5-tropic virus (**S3 Fig**), and viability was not reduced over 72 hours in culture with FTY720 (**S1A** and **S1C Fig**). In order to determine whether the inhibitory effect of FTY720 was due to an effect on infected (producer) cells, target cells, or both, we generated a co-culture system in which target CD4 $T_{CM}$ were cultured with infected (producer) $T_{CM}$. Target cells were pre-treated with 66nM FTY720 for 48 hours (the concentration used in cell-to-cell infection experiments), then labeled with Cell Trace Yellow dye immediately prior to co-culture with infected CD4 T cells. In parallel, untreated labelled targets were co-cultured with infected cells. Following 48 hours of co-culture, cells were harvested and infection assessed by flow cytometry. Intriguingly, in both donors we examined, the magnitude of the p24+ population in Cell Trace-labelled, pre-treated

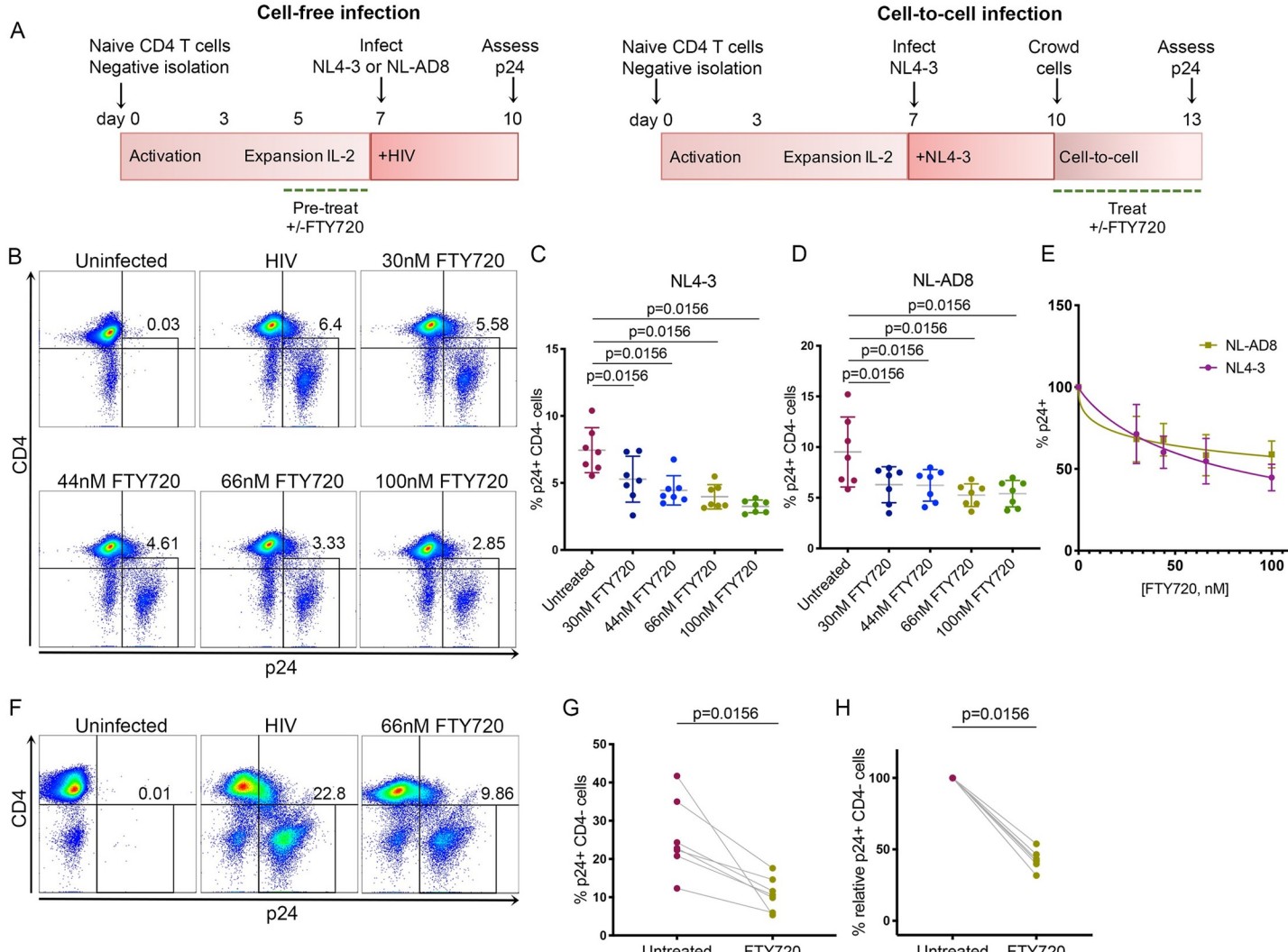

**Fig 1. Functional antagonism of S1P signaling inhibits HIV infection. A.** Primary $T_{CM}$ model of HIV infection (left: infection at day 7 with FTY720 pre-treatment; right: infection at day 7 followed by crowding and FTY720 treatment day 10–13). **B.** Representative flow cytometry plot of HIV-gag p24 expression at day 10 following pre- treatment with increasing doses of FTY720 (0-100nM) from days 5–7 and 3 days of culture post spin-infection with NL4-3 at day 7. **C.** Summary of 48-hr FTY720 pre-treatment followed by spin infection with NL4-3 and three days of culture. Mean + Standard Deviation (SD) is plotted. Data from seven individual donors (p = 0.0156 for Untreated vs 30, 44, 66 and 100nM FTY720, Wilcoxon matched-pairs signed-rank test) are shown. **D.** Summary of 48-hr FTY720 pre-treatment followed by spin infection with NL-AD8 and three days of culture. Mean + SD is plotted. Data from seven individual donors (p = 0.0156 for Untreated vs 30, 44, 66 and 100nM FTY720, Wilcoxon matched-pairs signed-rank test) are shown. **E.** Dose response curve for FTY720 with NL4-3 and NL-AD8 cell-free infection expressed as the percentage of infected cells (p24+) versus concentration of FTY720. **F.** Representative donor (of 7 total donors) from productive infection (day 13); uninfected, NL4-3 infected (no treatment), and NL4-3 infected (66nM FTY720 during days 10–13). **G.** Frequency of infected cells (p24+) at day 13 following treatment during crowding from days 10–13 +/- 66nM FTY720; NL4-3 infected (no treatment) and NL4-3 infected (FTY720 during crowding). Mean + SD is plotted. Data from seven individual donors are shown; p = 0.0156 by Wilcoxon matched- pairs signed- rank test. **H.** Relative infection day 13; NL4-3 infected (no treatment) and NL4-3 infected (FTY720 during crowding). Data are representative of seven individual donors; p = 0.0156 by Wilcoxon's matched- pairs signed- rank test. Normalization was performed as follows: % relative infection = $((d13_{treat}-d10_{treat})/(d13_{untr}-d10_{untr}))*100$.

target cells was approximately 1/3 that of labelled, untreated cells (2.52 vs. 6.26 and 8.11 vs 24.3% p24+, respectively (representative donor, **S4 Fig**). These results suggest that the inhibitory effect of FTY720 during cell-to-cell transmission is due to failure of HIV to infect the target cells rather than the inability of infected cells to produce virions. To verify this, we treated infected CD4 $T_{CM}$ with 66nM FTY720 and ART (1μM Raltegravir and 0.5μM Nelfinavir) for 72 hours in order to prevent further rounds of infection and assessed p24 (viral release) by

ELISA. We did not observe a difference in p24 production in the presence or absence of FTY720 during ART treatment, indicating that this S1P signaling modulator does not likely affect viral release in our model (**S5 Fig**).

## FTY720 reduces total and integrated HIV-1 DNA in T$_{CM}$

Using a primary cell model of latency, we have previously shown that latent infections are established during cell-to-cell transmission and the magnitude of latent infection directly correlates with productive infection measured at day 13 [34, 35]. Therefore, following our observation that FTY720 inhibits cell-free and cell-to-cell infection by HIV, we questioned whether pre-treatment with FTY720 (days 10–13) would lead to a reduction in levels of proviral DNA harbored within target cells and possibly to a reduction in the establishment of latency. To address this question, infected cells were treated with 66nM FTY720 from day 10 to 13. At day 13, FTY720 was removed from the culture and cells were cultured with 1μM Raltegravir and 0.5μM Nelfinavir from days 13 to 17 to block further viral spread. At day 17, we isolated the non-productively infected (CD4+) cells, which contain a mixture of uninfected and latently infected cells [34]. At this point, latent HIV infection was assessed by measuring total and integrated HIV DNA via nested PCR as previously described [36] (schematic, **Fig 2A**). We found that treatment with FTY720 resulted in a reduction in both total and integrated HIV DNA at day 17 (approximately 51.45+/-3.19 and 60.41+/-19.59%, respectively, **Fig 2B–2E**).

Next, we aimed to confirm that FTY720 interfered with the establishment of inducible latent HIV in CD4 T cells by assessing whether the reduction in the levels of HIV proviral DNA following FTY720 exposure translated to reduced reactivation of latent HIV. Latently infected cells were subjected to reactivation by TCR stimulation with αCD3/28 (or IL-2 only control) from days 17 to 19. TCR stimulation is one of the strongest stimuli that reactivates latent HIV in this model and in cells isolated from aviremic participants [35, 37]. As expected, the reduction of total and integrated HIV DNA due to FTY720 was accompanied by an average reduction in p24 following reactivation of 65.86+/-13.43% (**Fig 2F, 2G** and **2H**) relative to untreated controls. Interestingly, when FTY720 was added along with ART from days 13–17, there was no significant difference in virus reactivated from latency at day 19, indicating that in this model latent infection is established prior to ART and following ART exposure the latent reservoir is not impacted by FTY720 (**S6 Fig**). In light of our previous work demonstrating that in this model latently infected cells are established during cell-to-cell transmission, these results indicate that the brunt of the role of FTY720 on reduction of latency is likely a consequence of the reduction in productive infection. Overall, our results indicate that FTY720 not only reduces productive infection but that the effect is carried over into a reduced incidence of latently infected primary CD4 T cells.

## FTY720 reduces binding and fusion of HIV-1 in T$_{CM}$

Next, we wished to address the mechanisms by which FTY720 inhibits HIV infection. Our results demonstrating that FTY720 reduced both total and integrated HIV DNA suggested that FTY720 inhibits HIV infection at a step prior to reverse transcription. Therefore, we first examined whether there was a block to infection during viral binding or fusion. To quantify binding, T$_{CM}$ either pre-treated for 48 hours with 66nM FTY720 or untreated were incubated with NL4-3 (300ng p24) for 30 min at 4˚C, then immediately lysed for p24 assessment by ELISA (modification of [38, 39]). We observed that in FTY720 pre-treated T$_{CM}$ there was reduced virion binding in the majority of the donors tested (**Fig 3A**, 8 of 9 donors, average reduction 23.32+/-16.99%). Next, we examined viral fusion using NL4-3-BLaM. This virus carries beta-lactamase-Vpr chimeric protein (BLaM-Vpr) which permits determination of virion-

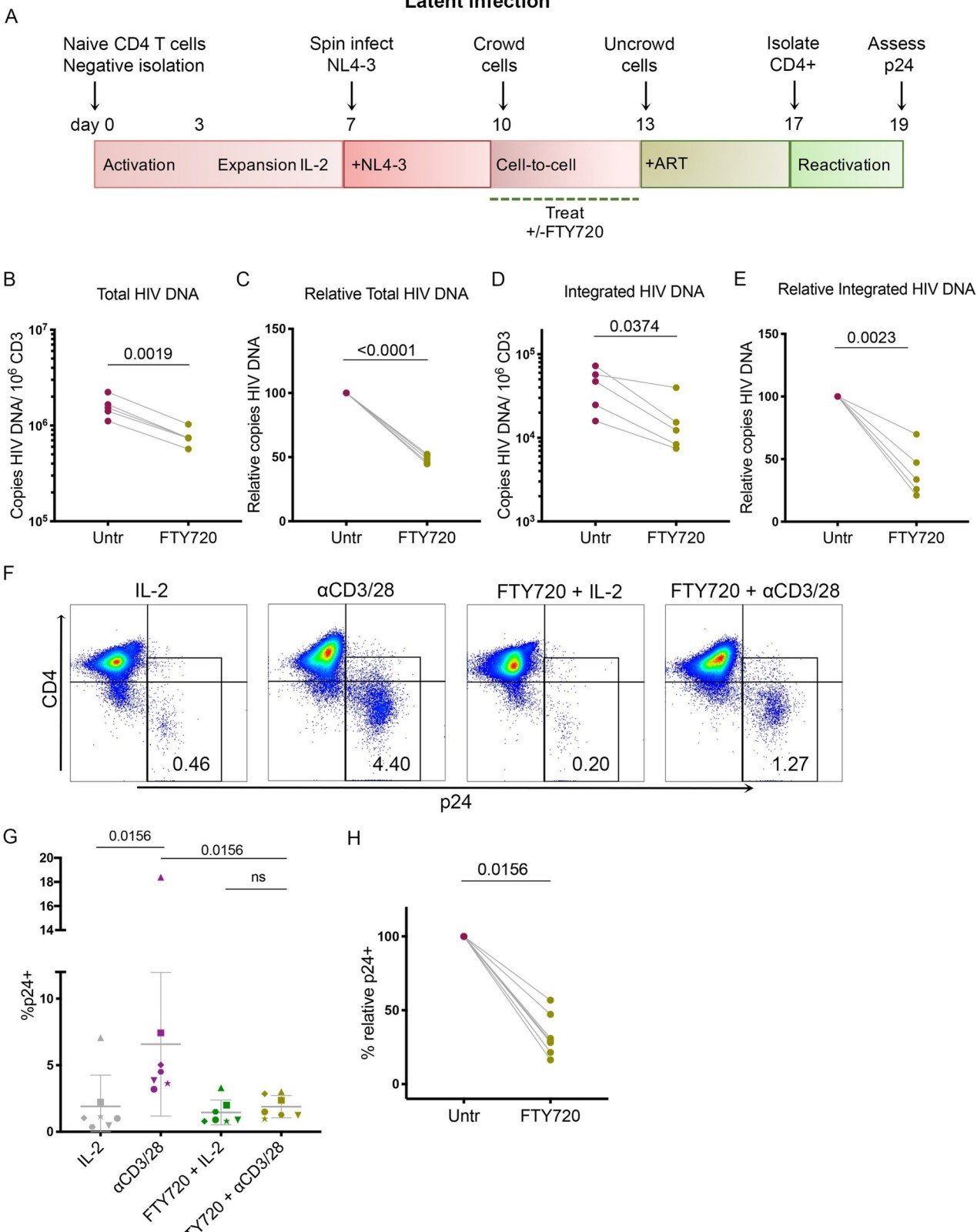

**Latent infection**

A

Naive CD4 T cells Negative isolation → Spin infect NL4-3 → Crowd cells → Uncrowd cells → Isolate CD4+ → Assess p24

day 0 — 3 — 7 — 10 — 13 — 17 — 19

Activation | Expansion IL-2 | +NL4-3 | Cell-to-cell | +ART | Reactivation

Treat +/-FTY720

B  Total HIV DNA — 0.0019

C  Relative Total HIV DNA — <0.0001

D  Integrated HIV DNA — 0.0374

E  Relative Integrated HIV DNA — 0.0023

F  IL-2 | αCD3/28 | FTY720 + IL-2 | FTY720 + αCD3/28
0.46 | 4.40 | 0.20 | 1.27

G  0.0156 | 0.0156 | ns

H  0.0156

**Fig 2. FTY720 treatment reduces the establishment of latency and total and integrated HIV-1 DNA in T$_{CM}$. A.** Schematic of primary cell model of HIV latency. For **B-E**, all figures represent nested PCR performed on CD4+ T cells isolated at day 17 with no treatment or FTY720 (66nM) treatment from day 10–13; whether total or integrated HIV DNA. Data are representative of five individual donors. **B.** Total copies of HIV DNA per 10$^6$ cells at day 17 (p = 0.0019, paired T-test). **C.** Relative total copies of HIV DNA per 10$^6$ cells (total copies of HIV DNA in treated/ total copies of HIV DNA in untreated) at day 17 (p<0.0001, paired T-test). **D.** Integrated copies of HIV DNA per 10$^6$ cells at day 17 (p = 0.0374, paired T-test). **E.** Relative integrated copies of HIV DNA per 10$^6$ cells (integrated copies of HIV DNA in treated/ integrated copies of HIV DNA in untreated) at day 17 (p = 0.0023, paired T-test). **F.** Representative donor from inducible latent HIV reactivation (day 19); untreated (IL-2 only), IL-2 + αCD3/28, IL-2 + FTY720 and FTY720 + αCD3/28. **G.** Frequency of infected cells (% p24+) at day 19 following CD3/28 stimulation days 17–19 with treatment during crowding from days 10–13 +/- 66nM FTY720; untreated (IL-2 only), IL-2 + αCD3/28, IL-2 + FTY720 and FTY720 + αCD3/28. Data is shown as Mean + Standard Deviation (SD). Data from seven individual donors, each with a unique symbol, are shown; p = 0.0156 for IL-2 only vs. αCD3/28 and αCD3/28 vs FTY720 + αCD3/28; p = ns for FTY720 + IL-2 vs FTY720 + αCD3/28, Wilcoxon matched-pairs signed-rank test for all comparisons. **H.** Relative % p24 upon reactivation by CD3/28 stimulation from days 17–19 (untreated or 66nM FTY720 treated from days 10 to 13, p = 0.0156, Wilcoxon matched-pairs signed-rank test). Normalization was performed as follows: % relative infection = ((d19$_{treat}$-d17$_{treat}$)/(d19$_{untr}$-d17$_{untr}$))*100.

host cell membrane fusion based on cleavage of the substrate CCF2 upon virus entry [40]. T$_{CM}$ pre-treated with FTY720 or untreated were incubated with NL4-3-BLaM and cleaved substrate was assessed by flow cytometry. As with binding, we observed that in the majority (7 of 8) of donors accessed, there was a reduction in viral fusion in FTY720 pre-treated T$_{CM}$ very similar to that of binding (**Fig 3B–3D**, average reduction 24.23+/-26.00%). As HIV binding and fusion depend on expression and proper aggregation of CD4 and the co-receptor CXCR4 or CCR5 [41–47], we assessed the expression of CD4 in cells either treated with FTY720 or untreated for 48 hours. We did not observe a difference in percent expression of CD4 following 48 hours of FTY720 exposure; however, we did observe a significant reduction in the mean fluorescence intensity (MFI) of CD4 (**Fig 3E**, average reduction 22.70+/-5.99%). Interestingly, the reduction in CD4 surface expression nearly matches that observed in both virion binding and fusion. There was not a statistically significant reduction in the MFI of CXCR4, but a significant reduction in the MFI of CCR5 on the CD4+ subset was observed (**Fig 3F** and **3G**, respectively). Thus, FTY720 pre-treatment of T$_{CM}$ results in a reduction of the surface density of CD4 that leads to a reduction in both binding and fusion. This reduction (average 23.52%) could only partially account for the strong reduction in integrated proviral HIV DNA observed (average 60.40%); thus, additional steps in the viral life cycle must be altered by FTY720 in CD4 T cells to account for the strong reduction in total and integrated HIV DNA.

## FTY720 promotes a reduction in phosphorylated SAMHD1 concomitant with a reduction in Cyclin D3

As S1P signaling modulators have previously been shown to cause cell cycle arrest in other model systems [48, 49] and quiescent T cells have been observed to be relatively non-permissive to infection by HIV-1 [1, 50], we investigated the effect of FTY720 on cell cycle state in our primary cell model. We treated uninfected T$_{CM}$ from day 10 of our model with 66nM FTY720 for 24, 48 and 72 hours and assessed cell cycle state by RNA/DNA staining (**Fig 4A**, one representative donor of five) in order to mimic as closely as possible the phenotype of the cells from our model used for cell-to-cell transmission experiments. Following FTY720 treatment, we observed an increase in G0 (**Fig 4B**) and G1b (**Fig 4C**) stages and a concomitant reduction in S/G2 phase (**Fig 4D**) starting from 24 hours post-treatment, which increased over time and was statistically significant at 48 and 72 hours (**Fig 4D**). In agreement with an increase in G0 and decrease in S/G2 phases, we also observed a reduction in the proliferation marker Ki67 at 48 hours post-treatment of uninfected cells with 66-100nM FTY720 (**Fig 4E and 4F**). The reduction in Ki67 with FTY720 treatment was also observed on infected cells (**S7 Fig**).

Following our observation that FTY720 promotes a non-cycling state in T$_{CM}$ and a reduction in total and integrated HIV DNA in our model, we hypothesized that FTY720 may be

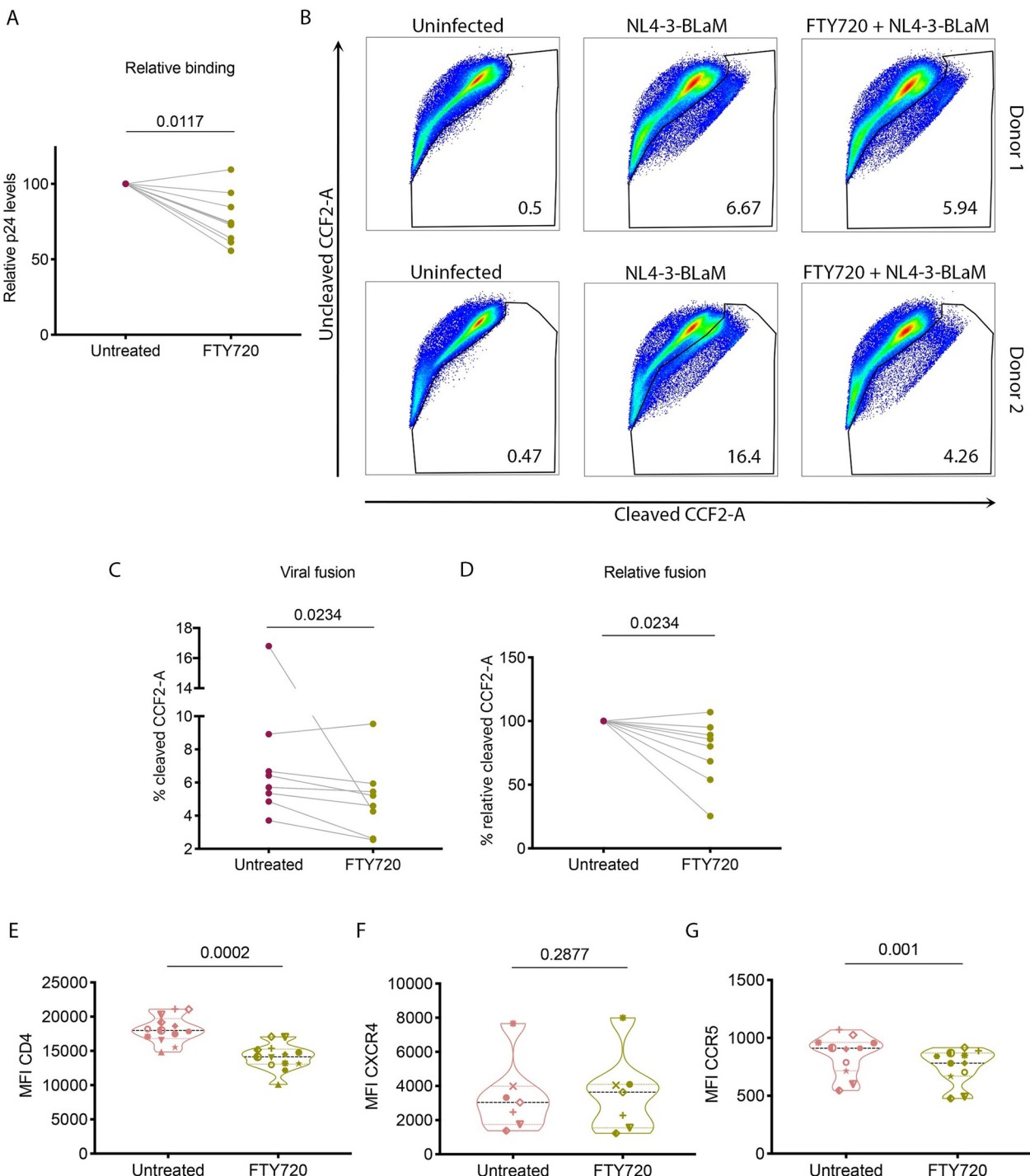

**Fig 3. FTY720 reduces binding and fusion of HIV-1 to $T_{CM}$. A.** Relative NL4-3 binding to $T_{CM}$ either untreated or pre-treated 48 hrs with 66nM FTY720 (n = 9, p = 0.0117 by Wilcoxon matched-pairs signed-rank test, treated values plotted as percent binding of untreated (pg/mL p24 in FTY720-treated/ pg/mL p24 in untreated)). **B.** NL4-3-BLaM fusion assay, two representative donors; uninfected, untreated NL4-3-BLaM infected and 66nM FTY720 + NL4-3-BLaM. **C.** Frequency of cells with HIV fusion events (%cleaved CCF2-A) in NL4-3-BLaM fusion assay (n = 8, p = 0.0234, Wilcoxon matched-pairs signed-rank test, raw values shown). **D.** Relative %cleaved CCF2-A for NL4-3-BLaM fusion assay (n = 8, p = 0.0234, Wilcoxon matched-pairs signed-rank test, treated values plotted as percent fusion events of untreated). **E.** Mean fluorescence intensity (MFI) of CD4 on CD4 $T_{CM}$ either untreated or pre-treated 48 hrs with 66nM FTY720 (n = 12, p = 0.0002, Wilcoxon matched- pairs signed- rank test, violin plot with median and quartiles shown). **F.** MFI of CXCR4 within CD4 $T_{CM}$ either untreated or pre-treated 48hrs with 66nM FTY720 (n = 7, p = 0.2877, Wilcoxon matched-pairs signed-rank test, violin plot with median and quartiles shown). **G.** MFI of CCR5 within CD4 $T_{CM}$ either untreated or pre-treated 48hrs with 66nM FTY720 (n = 11, p = 0.001, Wilcoxon matched-pairs signed-rank test, violin plot with median and quartiles shown). For **E**-**G**, each donor is represented by a unique symbol and these symbols denote the same donor across the three plots.

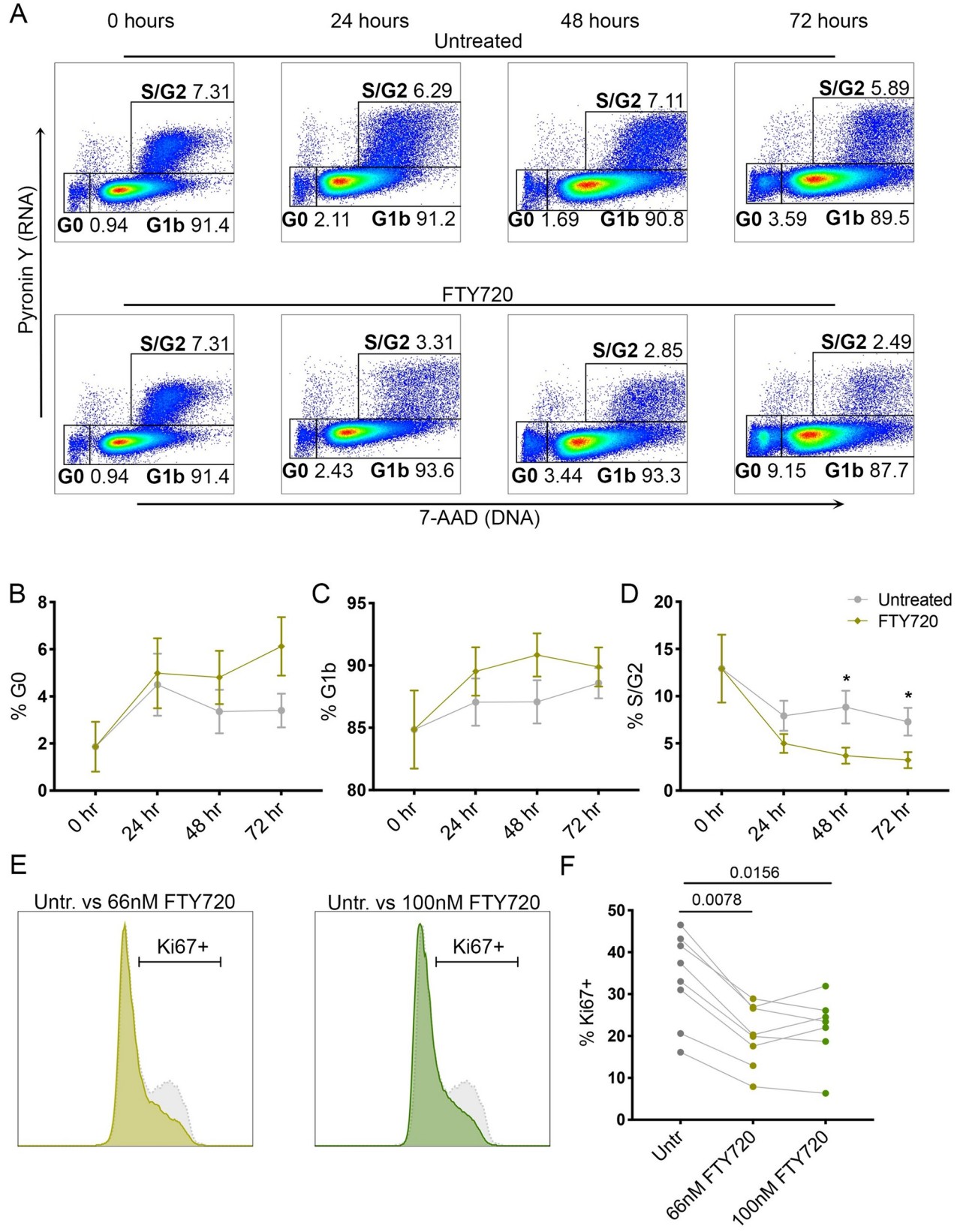

**Fig 4. FTY720 promotes a quiescent state and reduction in proliferation in T$_{CM}$. A.** RNA (PyroninY) and DNA (7-AAD) stain of T$_{CM}$ untreated or treated with FTY720 for 24, 48 and 72 hrs (one donor shown; representative of 5 donors). **B.** % G0 (PyroninY-/7-AAD-) T$_{CM}$ in untreated (grey) or FTY720 treated (green) T$_{CM}$ at 24, 48 and 72 hrs of culture (n = 5, Mean + Standard Error of Mean (SEM) shown). **C.** % G1b (PyroninY-/7-AAD +) in untreated (grey) or FTY720 treated (green) T$_{CM}$ at 24, 48 and 72 hrs of culture (n = 5, Mean + SEM shown). **D.** % S/G2 (PyroninY+/7-AAD+) in untreated (grey) or FTY720 treated (green) T$_{CM}$ at 24, 48 and 72 hrs of culture (n = 5, Mean + SEM shown, p = 0.028 for 48 hours and p = 0.044 for 72 hours by Multiple T test.). **E-F.** % Ki67+ cells in 48 hr untreated, 66nM FTY720 treated and 100nM FTY720 treated (all uninfected) T$_{CM}$ (days 10–12 of culture). **E.** Representative staining of Ki67 in untreated (grey dotted line) and 66nM FTY720 treated (light green), overlay of histograms; and Ki67 in untreated (grey dotted line) and 100nM FTY720 treated (forest green), overlay of histograms. One representative donor of 8 (66nM) or 7 (100nM) total donors. **F.** % Ki67+ cells in 48 hr untreated, 66nM FTY720 treated and 100nM FTY720 treated (all uninfected) T$_{CM}$ (days 10–12 of culture, p = 0.0078 and 0.0156 for 66 and 100nM, respectively, Wilcoxon matched-pairs signed-rank test, n = 7 or 8, Mean + SD shown).

promoting the activity of an innate HIV restriction factor. SAM Domain and HD domain-containing protein 1 (SAMHD1) is an established HIV-1 restriction factor that impairs reverse transcription via control of the dNTP pool and has been shown to be modulated by various cell cycle-related kinases in proliferating and non-proliferating cells [51, 52]. Thus, we examined the effect of FTY720 on expression of total and phosphorylated (inactive) SAMHD1. We treated T$_{CM}$ from day 10 of our model with 100nM FTY720 for 24 hours and quantified protein expression by Western blot. Following 24 hours of FTY720 treatment, we observed a significant reduction in pSAMHD1 (39.63+/-4.47%, **Fig 5A and 5B**) but did not observe an overall effect on total SAMHD1 levels (**Fig 5A and 5C**), indicating a relative increase in the active form of the restriction factor. SAMHD1 phosphorylation has been shown to be controlled by Cyclins and Cyclin-Dependent Kinases (CDKs) in primary human T cells and macrophages [53];[54]. Therefore, we evaluated whether FTY720 modulates the expression and phosphorylation of these kinases. We did not observe an effect on the levels of the majority of Cyclin-Dependent Kinases (CDKs) or Cyclins associated with transition from G1b to S phase of the cell cycle (CDK4, CDK6, Cyclin D2, CDK2, p21) or G2 to M phase (CDK1/pCDK1, **Fig 6A–6I**); however, we observed a significant reduction in the levels of Cyclin D3 (46.89 +/-14.61%, **Fig 6J**, all blots and respective β-actin controls shown in **S8 Fig**). Cyclin D3 has been demonstrated to regulate the activity of CDK6 and subsequently SAMHD1, the dNTP pool and HIV infection in macrophages [55]. Our results indicate a relative increase in the active (non-phosphorylated) form of SAMHD1, which may be associated with the decrease we observed in Cyclin D3. The mechanism of the regulation of SAMHD1 in our primary cell model remains to be elucidated.

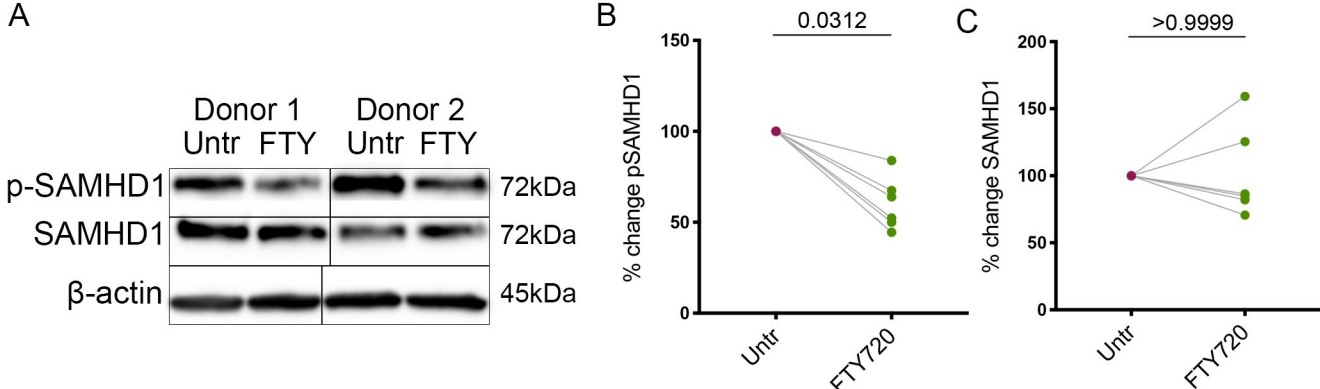

**Fig 5. Treatment with FTY720 induces a decrease in inactive (phosphorylated) SAMHD1. A.** Phosphorylated and total SAMHD1 and β-actin control for two representative donors (one male, one female), untreated and 100nM FTY720-treated, with molecular weight of each protein indicated. **B.** Phosphorylated SAMHD1 in 24 hr untreated and FTY720 treated cultures, p = 0.0312, Wilcoxon matched-pairs signed-rank test (n = 6). Phosphorylated SAMHD1 is normalized to β-actin control and pSAMHD1 of the treated is plotted as a percentage of that of the untreated. **C.** Total SAMHD1 in 24 hr untreated and FTY720 treated cultures, p>0.9999, Wilcoxon matched-pairs signed-rank test (n = 6). SAMHD1 is normalized to β-actin control and SAMHD1 of the treated is plotted as a percentage of that of the untreated.

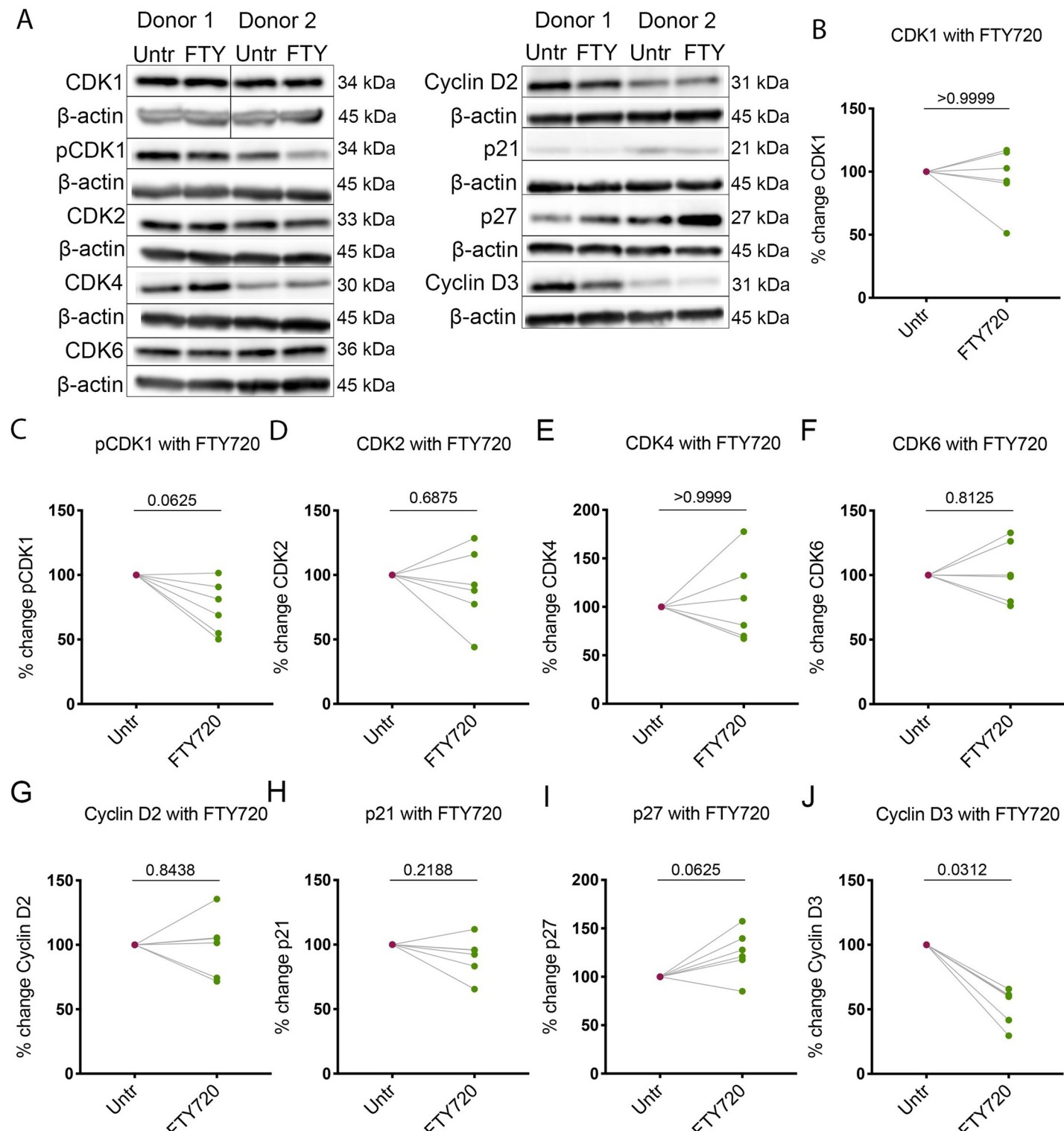

**Fig 6. Alterations in Cyclin D3 are observed concomitantly with pSAMHD1 reduction. A.** Representative Western Blot determination of protein levels of multiple Cyclin-Dependent kinases and Cyclins involved in cell cycle state, with molecular weight of each protein indicated. Shown are two representative donors of 6 total donors. **B-J.** Quantification of protein levels (by Western Blot) of the following Cyclin-Dependent Kinases, Cyclins or regulators of cell cycle, respectively: **B.** CDK1, **C.** pCDK1, **D.** CDK2, **E.** CDK4, **F.** CDK6, **G.** Cyclin D2, **H.** p21, **I.** p27, and **J.** Cyclin D3. p values displayed for Wilcoxon matched-pairs signed-rank test (all comparisons), n = 6 for all targets. For all proteins, the target is normalized to its respective β-actin control and expression of the treated is plotted as a percentage of that of the untreated.

## FTY720 does not impact the efficacy of previously characterized Latency Reversal Agents

An S1PR1 specific agonist, SEW2871, has previously been shown to be a latency-reversing agent (LRA) in an *in vitro* model of resting PBMCs, and FTY720 also demonstrated modest, albeit non-significant, LRA activity [14]. We therefore evaluated whether FTY720 could reactivate latent HIV alone or in combination with a panel of well-established LRAs with different mechanisms of action, including αCD3/28 (the positive control for latency reactivation in our model), the HDAC inhibitor SAHA (330nM), the TLR2/6 agonist Pam2CSK4 (1μM), the STAT SUMOylation inhibitor HODHBt (100μM) and the protein kinase C agonist Ingenol (100nM) [56–59]. We treated CD4 T cells isolated on day 17 of our model with 66nM FTY720 either alone or in combination with each LRA for 48 hours. The frequency of reactivated cells was measured by detection of the p24+ cells by flow cytometry. As can be seen in **Fig 7**, FTY720 did not hinder the ability of any of the LRAs tested to reactivate HIV from latency and showed

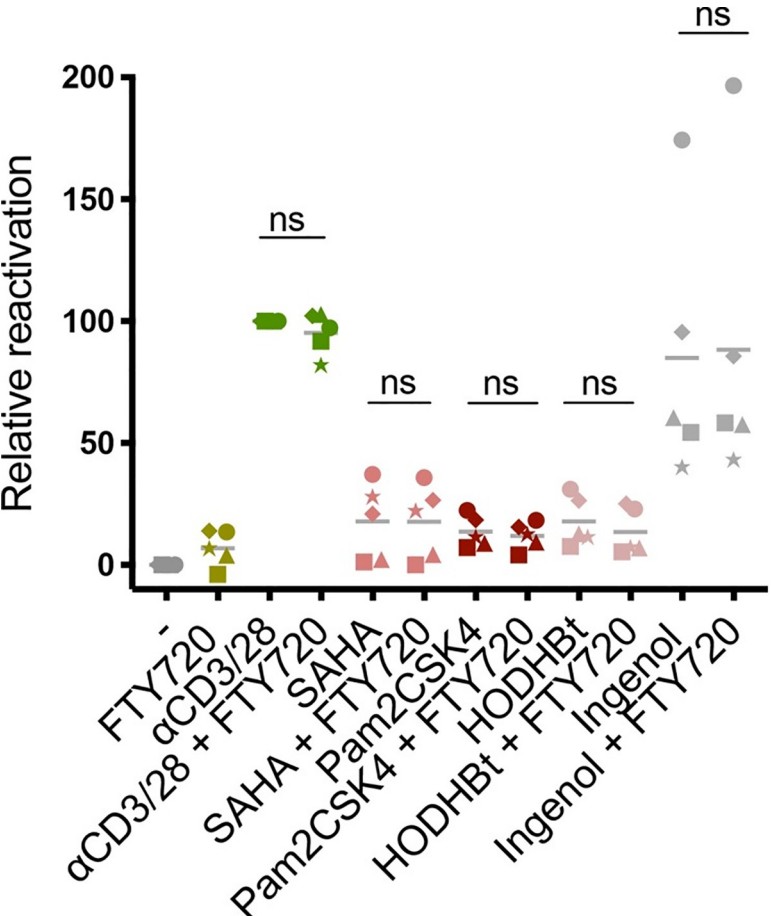

**Fig 7. FTY720 does not impact the efficacy of previously characterized Latency Reversal Agents.** CD4+ (non-productively infected) T_CM isolated at day 17 of our model of latency were cultured for 48hrs in the presence of 30 IU/mL IL-2 and a panel of several well-characterized Latency Reversal Agents (LRAs) alone or in combination with 66nM FTY720, including: the weak LRA/ control IL-2 only, αCD3/28 beads, SAHA (330nM), Pam2CSk4 (1μM), HODHBt (100μM) and Ingenol (100nM). Following the two days of culture each sample condition was accessed for the frequency of reactivated HIV (intracellular p24) by flow cytometry. For all LRAs, reactivation by each treatment is plotted as the percentage of the maximal stimulation in our model by αCD3/28, with each donor represented by a unique symbol (n = 5). Significance was determined by Wilcoxon matched- pairs signed- rank test for all comparisons.

moderate latency reversal activity alone in some of the donors tested. These results suggest that FTY720 could be a novel potential adjuvant in "Shock-and-kill" cure strategies as it does not interfere with the action of other LRAs and blocks further rounds of productive infection.

## Discussion

In this study, we investigated whether targeting S1P could inhibit the establishment of HIV-1 infection and the generation of the latent reservoir in CD4 T cells. Prior to our work, others had observed the potential of an S1P receptor agonist to reactivate HIV from latency as well as a decreased response to S1P signaling in chronically infected individuals [14, 31, 32]; however, the role of S1P signaling in establishment of infection and the potential to modulate this pathway to alter the course of infection or prevent establishment of the latent reservoir in CD4 T cells had not been reported.

Sphingolipids have been proposed to be involved in various stages of the HIV-1 life cycle [31, 60, 61]. Sphingolipids are integral components of the cellular membrane [62–66] and inhibition of glycosphingolipids strongly reduces HIV-1 fusion and productive infection in cell lines and primary T cells, respectively [31, 67]. Various groups have proposed a key role for this class of lipids in entry of HIV-1 in CD4 T cells via stabilization of the gp120-CD4 interaction necessary for fusion [68–71]. We took advantage of the well-characterized, clinically approved S1P receptor modulator FTY720/Fingolimod, currently utilized for treatment of MS, to examine the potential role of S1P-S1P receptor signaling in HIV infection as well as in the establishment of latency. Our results demonstrate that the effects of FTY720 in HIV infection are multifactorial. First, FTY720 reduces the surface density of CD4 on T cells, supporting a partial repression of virion binding and fusion by FTY720. Although we observed a reduction in the MFI of CCR5 on CD4 T cells, likely due to its reported co-expression with S1P receptor 1 in primary CD4 T cells [14], FTY720 inhibited infection irrespective of HIV-1 tropism, indicating a mechanism independent of coreceptor utilization. To our knowledge, there are no reports of CD4 and S1PR colocalization on CD4 T cells, although this could be a potential explanation of the reduction in MFI of CD4 observed. Alternately, the reduction in CD4 surface density could reflect the tendency that we observed of FTY720-treated cells to enter a resting state, or perhaps a destabilization of the cell membrane leading to less CD4 abundance, as glycosphingolipids have previously been implicated in other membrane interactions required for viral entry and fusion [68–71]. We propose a potential modulation by FTY720 of single or combined T cell surface features (CD4 abundance or distribution, lipid raft integrity, or stability of the viral synapse) resulting in the observed decrease in HIV binding and fusion. Our novel data indicate that there is yet much to uncover about the dynamics of membrane sphingolipids and HIV infection, as well as the potential regulation of membrane components by S1P modulators including FTY720.

As the blocks in viral binding and fusion together accounted for up to just 24% of the approximately 54% average reduction in productive HIV infection we observed, we examined additional stages of the HIV-1 life cycle to identify those which may be affected by modulation of S1P signaling. We measured the levels of total and integrated HIV-1 DNA by nested PCR and found that both forms were reduced (average of 51.45 and 60.41%, respectively) in FTY720-treated $T_{CM}$ relative to untreated $T_{CM}$. This result indicates that FTY720 inhibits infection at or prior to reverse transcription. This observation, along with our observation that $T_{CM}$ in our model, when treated with FTY720, were encouraged toward a resting phenotype, led us to examine the effect of FTY720 on innate cellular restriction factors that may be differentially expressed across cell cycle states and contribute to inhibition of HIV-1 infection. We discovered that the phosphorylated form of SAMHD1, a deoxynucleoside triphosphate

triphosphohydrolase that maintains cellular dNTP level balance and has also been shown to be an HIV-1 restriction factor [51, 52, 72] was significantly reduced by FTY720 treatment in $T_{CM}$ (average 39.6% reduction). Phosphorylated SAMHD1 is the inactive form of the restriction factor, thus, our observation of decreased levels of pSAMHD1 concomitant with maintenance of the total SAMHD1 availability indicates increased relative expression of the active form of this antiviral factor. SAMHD1 was first shown to restrict HIV-1 in myeloid and dendritic cells [51] and was recently demonstrated to have an analogous role in macrophages [73–75] and CD4 T cells [76–78]. In CD4 T cells, a cyclin-binding motif on SAMHD1 is required for its restriction of HIV [79] and its regulation by cell-cycle associated proteins including cyclins or Cyclin-Dependent Kinases (CDKs) has been demonstrated [54, 80]. In keeping with these reports, we observed a marked decrease in Cyclin D3 (average 46.98%) with FTY720 treatment along with increased relative expression of the active form of SAMHD1. Although Cyclin D3 has not yet been shown to interact with SAMHD1 in CD4 T cells as Cyclin A2 and Cyclin D2 have in other cell models [79, 81], this cyclin has been demonstrated to regulate SAMHD1 and subsequently permissiveness to HIV infection in macrophages [55] and its expression was altered by FTY720 in a model of murine diabetes [82]. As we also observed a reduction in S/G2 cell cycle phase and Ki67 expression in $T_{CM}$ treated with FTY720, our results are consistent with a model of an FTY720-induced decrease in cell cycling along with a reduction of Cyclin D3 and potentially increased activity of SAMHD1. We therefore propose that Cyclin D3 may be a regulator of SAMHD1 activity in a cell-cycle dependent manner and that S1P signaling may regulate the levels of Cyclin D3 in CD4 T cells. Additional investigation will be needed to confirm this and determine whether this is a direct regulation by phosphorylation or whether other players are involved. Alternately, a different kinase may regulate activity of SAMHD1 via phosphorylation in $T_{CM}$ and this kinase is inhibited by FTY720 in CD4 T cells.

Moreover, we aimed to determine the extent to which the effect of FTY720 observed during cell-free and cell-to-cell infection would impact reactivation of latent HIV. We observed a significant decrease in virus reactivated from latency in cells cultured with FTY720 from days 10 to 13, indicating that the ability of FTY720 to decrease infection translated to an inhibition of seeding of the latent reservoir. In fact, the decrease in latency was more potent than the effect on both cell-free and cell-to-cell infection (65.86% latency reduction vs 45.34 and 54.4% cell-free and cell-to-cell at 66nM FTY720, respectively). This decrease in infection level was supported by nested PCR results, which revealed a 60.4% reduction in integrated DNA. Thus, the reduction observed in integrated virus and the inhibition in the level of reactivation occur to approximately the same extent in our model. This suggests that FTY720 strongly reduces establishment of latency. Recently, Abrahams *et al* demonstrated that most latently infected cells are generated near the time of ART initiation [83]. When we treated cells in our model with FTY720 concomitantly with ART, we readily observed reactivation of HIV, albeit no difference in FTY720-treated or untreated samples. Therefore, our results suggest that FTY720 may be useful as a strategy to limit the size of the latent reservoir if used prior to ART initiation, such as in acute infection. Finally, we examined whether FTY720 would interfere with the ability of a panel of established LRAs to reactivate latent HIV in our model. A previous report by Duquenne *et al* indicated that an S1PR1 agonist, SEW2871, could induce viral production in an *in vitro* model of HIV latency in resting PBMCs [14]; however, in our primary cell model of latent HIV infection, we observed only weak LRA activity of FTY720 following 48 hours of culture, likely due to differences in the models used as well as differing mechanisms of FTY720 and other S1PR1 antagonists/ agonists. Importantly, we did not observe an impairment of the activity of several well-established LRAs in the presence of FTY720, indicating that FTY720 may be used in combination with other LRAs towards reducing the latent HIV reservoir.

In conclusion, our results indicate that FTY720 may be an exciting novel therapy for HIV infection. FTY720 is already clinically approved and well-tolerated and we show that it also restricts HIV infection of CD4 T cells. As such, targeting the S1P-S1PR axis may be an alternative strategy to employ in the context of prevention as a potential microbicide; as an adjuvant to current ART strategies to reduce the seeding of the latent reservoir; or in strategies aimed toward viral reactivation and eradication of the latent reservoir.

## Materials and methods

### Ethics statement

Cells were isolated from buffy coats of anonymous healthy blood donors obtained from the Gulf Coast Regional Blood Center (GCRBC) in Houston, Texas. GCRBC provided us with only the biological sex and age of donors. No other personal identifiable information (including race) was provided.

### Reagents

The following reagents were provided by the AIDS Research and Reference Reagent Program, Division of AIDS, National Institute of Allergy and Infectious Diseases (NIAID, MD): Nelfinavir (cat. # 4621), Raltegravir (cat. # 11680) from Merck & Company (NJ) and HIV-1$_{NLAD8}$ and HIV-1$_{NL4-3}$ from Malcolm Martin (MD, cat. #s 11346 and 114). Human αIL-12 and αIL-4 were purchased through PeproTech (NJ). Human rIL-2 was obtained through the BRB/NCI Preclinical Repository (MD). Antibodies were purchased from BD (NJ, Kc57-FITC, CD4-APC), Biolegend (CA, Ki67), and eBiosciences (CA, eF450 fixable viability dye, CXCR4-PE-Cy5.5, CCR5-AF488). FTY720 (Fingolimod) was obtained from Cayman Chemical (MI, cat. # 10006292, CAS#162359-56-0).

### Cell culture and generation of primary cell model of latency

Human peripheral blood mononuclear cells were obtained from healthy, unidentified blood donors (Gulf Coast Regional Blood Center (GCRBC), TX). Naive CD4 T cells were isolated from PBMCs by negative selection and activated in non-polarizing conditions at $0.5x10^6$ cells/mL in the presence of 2μg/mL αhuman IL-12, 1μg/mL αhuman IL-4, 10ng/mL TGF-β and αCD3/28 stimulation beads at one bead/cell (Dynal/Invitrogen, CA) as previously performed [33, 34]. Subsequently, cells were expanded with 30 IU/mL hIL-2 in RPMI supplemented with 1% L-Glutamine, 10% Fetal Bovine Serum and 1% Penicillin/Streptomycin. Incubation with FTY720 was performed on cells from various time points of our model for 24–72 hours at concentrations of 30-100nM. HIV-1$_{NLAD8}$ and HIV-1$_{NL4-3}$ viruses were generated in HEK293FT cells by calcium phosphate transfection and latently infected cells were generated as previously described [33, 34].

For latency reversal experiments, cells were cultured at $0.5x10^6$ cells/mL in the presence of 30 IU/mL hIL-2. Reactivation conditions included: IL-2 only, 66nM FTY720, or one of the following LRAs +/- 66nM FTY720: αCD3/28 stimulation beads (one bead/cell), SAHA (330nM, Cayman), Pam2CSK4 (1μM, Invivogen), HODHBt (100μM, A.K. Scientific), or Ingenol (100nM, Cayman). Following 48 hours of culture, intracellular p24-gag was assessed by flow cytometry.

### Flow cytometry

To analyze productively infected cells, $2.5x10^5$ cells were stained for CD4 (clone: S3.5, APC, BD), fixable viability dye (eF450, eBiosciences) and intracellular p24-gag (BD) and were

analyzed on a Celesta flow cytometer (BD) as previously performed [34]. For proliferation studies, cells were stained for CD4, viability and intracellular Ki67 (clone: 6604665, FITC, Biolegend). To analyze cellular RNA/ DNA content in order to determine cell cycle state, cells were stained with 7-AAD (Millipore Sigma, MA) and PyroninY (Sigma, MO) as previously done [84]. To label cells prior to co-culture assays, cells were stained with Cell Trace Yellow proliferation dye (ThermoFisher, MA). Flow cytometry analysis was performed in FlowJo software (BD) and further statistical analysis was performed in GraphPad Prism (CA).

## Quantitative real-time PCR for total and integrated HIV DNA

Nested PCR was performed as previously described [36]. Briefly, CD4 T cells were digested with Proteinase K. A first round of PCR pre-amplification (12 cycles) was performed directly on cell lysates using primers for the LTR/*gag* region (total HIV DNA) or primers specific for the LTR region (ULF1) together with two *Alu* primers (integrated HIV DNA). A nested real-time PCR was then carried out on a Rotor-Gene Q instrument (Qiagen, Mississauga, Canada) using inner primers and a TaqMan probe. The number of copies of the *CD3* gene was determined to accurately quantify the number of cells in each reaction. Results were expressed as HIV DNA copies per $10^6$ CD3 T cells.

## NL4-3-BLaM assay

$T_{CM}$ were cultured for three hours with NL4-3-BLaM to determine viral fusion with the host cell membrane by incubation with beta-lactamase-Vpr chimeric proteins (BLaM-Vpr) substrate and flow cytometry as previously done [40].

## Protein analysis

To analyze protein expression, $10^7$ cells/mL were lysed in NETN lysis buffer (100mM NaCl, 20mM Tris-Cl (pH 8.0), 0.5mM EDTA, and 0.5% v/v Nonidet P-40 (NP40) supplemented with protease and phosphatase inhibitors (cOmplete mini, Millipore Sigma and PhosStop, Roche, Basel, Switzerland, respectively). Protein concentration was quantified by BCA assay and 10μg of both untreated and 100nM FTY720 pre-treated (24hr) sample were subjected to gel electrophoresis. Following transfer to nitrocellulose membrane, membranes were blotted with antibodies to phosphorylated and unphosphorylated SAMHD1 (cat. # 89930S, cat. # 12361), CDK1 (cat. # 77095), pCDK1 (cat. # 4539), CDK2 (cat. # 2546), CDK4 (cat. # 12790), CDK6 (cat. # 3136), Cyclin D2 (cat. # 3741), Cyclin D3 (cat. # 2936), p27 (cat. # 3686), and p21 (cat. # 2947) plus β-actin control (cat. # A5441), all from Cell Signaling Technologies (MA), followed by HRP-conjugated secondary antibody (cat. #s 115-035-146 (α-mouse) and 111-035-046 (α-rabbit), both Jackson Immunoresearch, PA) and visualization with Immobilon HRP substrate (Millipore) and GeneGnome software (Syngene, Bangalore, India). Quantification was performed in GeneTools (Syngene).

## Supporting information

**S1 Fig. Viability of FTY720-treated $T_{CM}$ A.** $T_{CM}$ obtained by immunomagnetic isolation from PBMCs were cultured with 66nM FTY720 for 72hrs and viability was evaluated by fixable viability dye and Activated Caspase 3 staining. Representative donor staining of viability dye and activated Caspase 3 on $T_{CM}$. **B-C.** $T_{CM}$ obtained from naïve cells expanded in our primary cell model of latency were infected with NL4-3 or NL-AD8 and pre-treated +/-FTY720 (30–100 nM) and stained for flow cytometric assessment of viability at day 10 or 13. **B.** Viability of 7 donors infected with NL4-3 and 7 donors infected with NL-AD8 pre-treated +/-30-100nM

FTY720 from day 5–7 and infected from day 7–10 (as in **Fig 1A** left schematic). **C.** Viability of 7 total donors infected with NL4-3 at day 7 (as in **Fig 1A** right schematic) and treated+/- 66nM FTY720 from day 10–13. For **B-C.**, Wilcoxon signed-rank matched-paired tests were used for all comparisons.
(TIF)

**S2 Fig. Infection of CD4 T cells with HIV 89.6 and JR-CSF** Primary CD4 T cells expanded and pre-treated at day 5 with 66 or 100nM FTY720 (as in **Fig 1A**, left schematic) were infected with dual-tropic HIV-1 (89.6) or R5-tropic HIV-1 (JR-CSF) at day 7 of culture. Frequency of p24+ cells was assessed at day 10 by flow cytometry (N = 3 donors for each virus and each concentration of FTY720). **A.** Representative donor infected with 89.6 and JR-CSF, either untreated or treated with two concentrations of FTY720. **B.** Summary of infections with 89.6 and JR-CSF. Data are expressed as the percent of infection in the FTY720-treated conditions relative to untreated. Mean + SD are shown; statistical comparison was performed by paired T-test and is color coded for each virus (green = JR-CSF, purple = 89.6).
(TIF)

**S3 Fig. Functional antagonism of S1P signaling inhibits cell-to-cell transmission of R5-tropic HIV-1.** CD4 T cells were infected at day 7 with NL-AD8, crowded and treated with 66nM FTY720 from day 10–13, and assessed for frequency of infected cells by flow cytometry. **A.** Two representative donors from productive infection (day 13); uninfected, NL-AD8 infected (no treatment), and NL-AD8 infected (66nM FTY720 from day 10–13). **B.** Schematic of the experimental design. **C.** %p24+ cells at day 13 following treatment during crowding from day 10–13 with (or without) 66nM FTY720. Data comprise four total donors, each represented by a unique symbol. Statistical comparison was performed by paired T-test.
(TIF)

**S4 Fig. Co-culture of labeled FTY720-treated and NL4-3 infected $T_{CM}$.** $T_{CM}$ either treated or untreated with 66nM FTY720 for 48 hrs were labeled with Cell Trace Yellow dye and co-cultured with unlabeled pre-crowded NL4-3 infected (producer) $T_{CM}$. 48hrs later, pre-treated and untreated target cells were evaluated for intracellular expression of p24 by flow cytometry, gating on Cell Trace Yellow+ cells. Shown is one representative donor of two individual donors (uninfected, untreated, and FTY720 pre-treated Cell Trace-labeled target cells and unlabeled producer cells.)
(TIF)

**S5 Fig. FTY720 treatment during ART does not alter viral release.** Primary CD4 T cells from our model were infected at day 7 and treated at day 10 with ART+/- 66 or 100nM FTY720, followed by assessment of p24-gag by ELISA at day 13 in order to determine the effect of FTY720 on viral release. **A.** Schematic of p24 ELISA following ART+/-FTY720 for 72 hours (days 10–13). **B.** Summary of p24 ELISA at day 13 following treatment of infected cells from day 10–13 with ART+/-FTY720 (either untreated or +FTY720, n = 4, statistical comparisons: paired T-test).
(TIF)

**S6 Fig. FTY720 treatment during ART does not alter reactivation from latency** Primary CD4 T cells from our model of HIV latency were infected with NL4-3 at day 7, crowded at day 10, uncrowded at day 13 and treated for 4 days with ART (1μM Raltegravir/ 0.5 μM Nelfinavir) in the presence or absence of 66nM FTY720 prior to isolation of non-productively infected (CD4+) cells at day 17 and reactivation of latent HIV-1 for 48 hours with αCD3/28 or IL-2 only control. Frequency of reactivated virus (%p24+ cells) was assessed by flow cytometry at

day 19. **A.** Schematic of latency reversal following FTY720 treatment during ART. **B.** Two representative donors (of 4 individual donors) from day 19, following 48 hours of reactivation with αCD3/28 or IL-2 only control, either untreated or +66nM FTY720 from day 13–17. **C.** Summary of day 19 reactivation with αCD3/28 or IL-2 only control (either untreated or +66nM FTY720 from day 13–17, n = 4, both statistical comparisons: paired T-test). Mean is indicated and each donor is represented by a unique symbol.
(TIF)

**S7 Fig. Ki67 Expression on NL4-3 infected cells treated with FTY720.** Primary CD4 T cells were cultured and expanded, infected with NL4-3 at day 7, crowded at day 10 and treated or not treated with 66nM FTY720, and stained at day 13 for flow cytometry to assess the expression of the proliferation marker Ki67. Two infected donors were stained. Grey dotted line: uninfected/ untreated; green dotted line with light fill: uninfected + 66nM FTY720; pink dotted line: NL4-3 infected/ untreated; dark green filled histogram: NL4-3 infected + 66nM FTY720.
(TIF)

**S8 Fig. Western Blots of cell cycle proteins and β-actin controls.** Primary CD4 T cells (Cultured T Central Memory cells) were treated for 24 hours +/- 100nM FTY720 and were lysed for Western Blot determination of protein levels of multiple Cyclin-Dependent kinases and Cyclins. Shown are 6 total donors assayed for: CDK1, pCDK1, CDK2, CDK4, CDK6, Cyclin D2, p21, p27, Cyclin D3, pSAMHD1 and total SAMHD1, with the molecular weight of each protein indicated.
(TIF)

## Author Contributions

**Conceptualization:** Rachel S. Resop, Alberto Bosque.

**Formal analysis:** Rachel S. Resop.

**Funding acquisition:** Alberto Bosque.

**Investigation:** Rachel S. Resop, Daniel Newman, Hawley Rigsby, Larisa Dubrovsky.

**Methodology:** Rachel S. Resop, Alberto Bosque.

**Supervision:** Rémi Fromentin, Michael Bukrinsky, Nicolas Chomont.

**Visualization:** Rachel S. Resop.

**Writing – original draft:** Rachel S. Resop.

**Writing – review & editing:** Rachel S. Resop, Michael Bukrinsky, Nicolas Chomont, Alberto Bosque.

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
