## [Decision Letter · Decision Letter 0]

1 Feb 2020

Dear PhD Bosque,

Thank you very much for submitting your manuscript "Fingolimod inhibits multiple stages of the HIV-1 life cycle" for consideration at PLOS Pathogens. As with all papers reviewed by the journal, your manuscript was reviewed by members of the editorial board and by three independent reviewers. In light of the reviews (below this email), we would like to invite the resubmission of a significantly-revised version that takes into account the reviewers' comments.

We cannot make any decision about publication until we have seen the revised manuscript and your response to the reviewers' comments. Your revised manuscript is also likely to be sent to reviewers for further evaluation.

Sincerely,

Daniel C. Douek

Associate Editor

PLOS Pathogens

Susan Ross

Section Editor

PLOS Pathogens

Kasturi Haldar

Editor-in-Chief

PLOS Pathogens

orcid.org/0000-0001-5065-158X

Michael Malim

Editor-in-Chief

PLOS Pathogens

orcid.org/0000-0002-7699-2064

Reviewer's Responses to Questions

**Part I - Summary**

Reviewer #1: The authors aim to describe the molecular mechanism through which FTY720 inhibits the viral life cycle. Unfortunately, in my view, the effects are weak and not very conclusive. Furthermore, while the use of primary cells is appreciated, in some figures (eg: 3B and D), this induces huge variability. I would recommend testing some of the hypothesis cited in the manuscript on additional models of HIV-1 infection, including cell lines and additional viral strains.

Reviewer #2: Resop and colleagues study how Fingolimod (FTY720), a modulator of the S1P pathway, impacts HIV-1 replication. In an overall well-conducted, convincing and well-presented study, they show that FTY720 inhibits both cell-free and cell-to-cell infection. They demonstrate that this is due to multiple effects of the compound: downregulation of surface CD4 and CCR5, lower proliferative capacities and a higher relative expression of activated SAMHD1 (less pSAMHD1), likely due to the downregulation of CDK3, although no mechanistic link between the two is presented here. They further show that treatment with FTY720 leads to fewer latently infected cells, leading to fewer p24-expressing cells following reactivation. This is however expected as FTY720 decreases the levels of productively infected cells, a proportion of which becomes latently infected. They finally look at the latency reversal activity of FTY720, which only displays a poor effect by itself (if any), and doesn’t modify the activity of the other latency reversal agents tested, which strengthen its potential role for shock and kill strategies. This interesting work is in line with previous studies examining the role of SP1R modulation during HIV-1, with a more virological and mechanistic point of view. Overall, this study provides evidence for the potential use of a FDA-approved drugs for HIV-1 treatment.

Reviewer #3: In the manuscript, “Fingolimod inhibits multiple stages of the HIV-1 life cycle” Resop and colleagues investigated the effect of FTY720, a sphingosine-1-phosphate receptor antagonist on HIV infection. The authors reported that FTY720 is able to block HIV infection, by inhibiting viral binding and fusion and by increasing the activity of the viral restriction factor SAMHD1. This is a very well-written manuscript and highlights a novel target that could potentially be exploited to inhibit HIV replication. That said, the manuscript suffers from inconsistencies and made some claims that were not completely supported by the data presented. The strength of this manuscript can be significantly improved by addressing weaknesses outlined below.

1. There is a general lack of important details with many of the figures. For example, in all figures with scatter plots, it is not really clear what is plotted. Mean + SD? Mean + range? How is the data normalized when relative percent are shown? This is especially important for the western blot figures 5 and 6. Is the data standardized to the loading control as it should be?

2. The authors stated that "FTY720 did not alter viability at any of the concentrations tested" (line 129), but figure S1 only shows one concentration (66nM). Viability should be shown for all the concentrations.

3. An important control for figure 2 should be an inclusion of viability data following treatment for 3 days with FTY720 to show that the reduction in HIV DNA is not due to cell death. Viability data in figure S1 is from 48hrs.

4. Line 236, “we observed a significant reduction in pSAMHD1 (39.63 +/- 14.47%, Figs 5A and 5B) but did not observe an effect on total SAMHD1 levels (Figs 5C and 5D)”. Can the authors comment on the fact that it appears that there are changes in total SAMHD1 levels (figure 5C), but perhaps inconsistent across the donors?

5. Line 247, the authors state that that their results “results suggest a model of regulation of CDK4/6 by Cyclin D3, which in turn results in a decrease in phosphorylation and corresponding activation of the activity of the restriction factor SAMHD1” However, they did not really provide direct evidence of SAMHD1 activation. A functional assay showing SAMHD1 activation would be necessary to support their claim.

6. Given that FTY720 is an immunomodulatory agent, and their data shows an effect on the cell cycle, including an increase towards a more resting, G0 phenotype following FTY720 treatment, can the authors comment on 1) the status of activation of their Tcm and how that might influence some of their results, including a limit in HIV infection and 2) the level of transcription factors necessary for efficient HIV transcription (e.g . P-TEFb) in cells treated with FTY720 and how that might also affect their observations on HIV infection ?

**Part II – Major Issues: Key Experiments Required for Acceptance**

Reviewer #1: To validate their data, the authors should perform experiments in other models fo HIV infeciton, including either cell lines or additionnal virla stratins.

Reviewer #2: Main concerns

1) Figure 1: is there a control for the activity of FTY720 on SP1R? Which SP1R are important for the effects observed? At least, the authors should discuss which SP1R are expressed on CD4 T cells and how HIV-1 infection modulates their expression.

2) In figure S1, it would be informative to show a graph with the viability for all donors and doses, especially the highest tested (100 nM).

3) Figure 1 panel H should be placed before E, F and G as it is mentioned before in the text and concerns cell-free infection (which should be indicated in the legend).

4) Figure 1 panel A schematic can be confusing and lead to the misleading interpretation that cells used for the cell-to-cell assay were used for the cell-free assay and treated twice with FTY720. Maybe splitting the schematic into two (one for cell-to-cell and one for cell-free) would be clearer. Moreover, as HIV-1 reactivation is assessed later in the manuscript, the experimental design could be rather added as a schematic in figure 2.

5) For the cell-to-cell results of Figure 1E-G and Figure S2, showing the Gag/CD4 staining in the unlabeled producer cells would help strengthening the conclusion that FTY720 is mainly acting on target cells susceptibility to infection. Moreover, to study the effect of FTY720 on producer cells, infected cells could be cultivated in presence of ART +/- FTY720 and viral production in the supernatant assessed by a p24 ELISA. Are the effects on cell-to-cell transmission also independent of viral tropism?

6) How many donors are the results presented in Figure S2 representative of? This should be added in the legend.

7) The results presented in Figure 2 are somewhat expected. Indeed, it was shown in Figure 1 that the levels of infection at day 13 are lower when cells are treated with FTY720 during crowding. Thus, it is expected that after 4 days of ART, fewer cells will harbor HIV-1 DNA. Adding the compound during the ART treatment might help examining whether FTY720 modulates the seeding of the reservoir per se by examining the proportion of infected cells that will become latently infected.

8) In panels A-D of figure 2, it seems surprising to reach p values of <0.0001 with a Wilcoxon test on 5 donors.

9) In panels A-D of figure 2, indicating whether we are looking at total or integrated DNA would make the figure clearer.

10) Do they authors have any data or hypotheses for the mechanisms of downregulation of CD4 presented in Figure 3. This could be at least discussed.

11) In figure 4, why did the authors wait until day 10 for treatment with FTY720?

12) In figure 4 panels B, C and D, adding a legend (grey, green) on the figure would improve the readability and performing statistical analysis would strengthen the interpretation.

13) Relative to Figure 4, in the context of infection, does treatment with FTY720 impact the cell cycle or proliferation of infected cells?

14) To really look at the importance of SAMHD1 in the effects of FTY720, the authors could repeat the experiments in cells treated with VLP-Vpx. Looking at whether HIV-2 replication, which counteracts SAMHD1, is less inhibited by FTY720 could be of interest in a future study

15) Panel 5D is redundant with 5B and 5C.

16) Panel 6G title should be Cyclin D2 (space missing) and 6H should be p21 (not p27).

17) Results presented in Figure 7 suggest that FTY720 is rather inactive for latency reversal. Repeating these experiments in ART-treated patients’ samples would strengthen the conclusions on the role of FTY720 during latency reversal. Moreover, how may FTY720 modulate CD4 T cell activation? This could be discussed

18) In the discussion, line 323, it is stated that there is an “increased expression of active SAMHD1”. This should be changed to “relative expression” as what was observed is a lower expression of the inactive phosphorylated form.

19) In the Methods section, line 409 should read “Results were expressed as HIV DNA copies per 106 CD3 T cells”.

Reviewer #3: 1. An important control for figure 2 should be an inclusion of viability data following treatment for 3 days with FTY720 to show that the reduction in HIV DNA is not due to cell death. Viability data in figure S1 is from 48hrs.

2. Viability data for all concentrations of FTY720 should be included.

3. The authors need to provide direct evidence to support their claim that FTY720 treatment results in an increase of activated SAMHD-1

**Part III – Minor Issues: Editorial and Data Presentation Modifications**

Reviewer #1: The viral titer should be quantified in some experiments (eg: Fig1) to show if the treatments affect virus production.

In Fig. 2: absolute number of copies should be presented.

In Fig. 3D: there is no visible change on the BLAM ssay for at least donor 1

In Fig. 4: show cell cycle profile

Reviewer #2: (No Response)

Reviewer #3: 1. Minor, but confusing, figure 1 in text not in sequence. 1B, 1C , 1D then 1H. Should be fixed.

2. In figures with scatter, it would make it easier for the reader to evaluate the response across donors if a unique symbol is used for each donor.

3. As western blot densitometry tracing is only semi-quantitative, the blots for all the donors should be included as a supplementary figure to better help the reader evaluate the data.

PLOS authors have the option to publish the peer review history of their article (what does this mean?). If published, this will include your full peer review and any attached files.

Reviewer #1: No

Reviewer #2: No

Reviewer #3: No
---

## [Decision Letter · Decision Letter 1]

23 May 2020

Dear PhD Bosque,

Thank you very much for submitting your manuscript "Fingolimod inhibits multiple stages of the HIV-1 life cycle" for consideration at PLOS Pathogens. As with all papers reviewed by the journal, your manuscript was reviewed by members of the editorial board and by several independent reviewers. The reviewers appreciated the attention to an important topic. Based on the reviews, we are likely to accept this manuscript for publication, providing that you modify the manuscript according to the review recommendations. One of the reviewers has a few remaining very minor concerns which you should able to address easily.

Sincerely,

Daniel C. Douek

Associate Editor

PLOS Pathogens

Susan Ross

Section Editor

PLOS Pathogens

Kasturi Haldar

Editor-in-Chief

PLOS Pathogens

orcid.org/0000-0001-5065-158X

Michael Malim

Editor-in-Chief

PLOS Pathogens

orcid.org/0000-0002-7699-2064

Reviewer Comments (if any, and for reference):

Reviewer's Responses to Questions

**Part I - Summary**

Reviewer #2: The authors have taken into account my concerns and the manuscript has been strongly improved. The addition of more background about FTY720 and SP1R is appreciated, as well as the addition of experiments suggested by the reviewers. These novel results, together with modifications of text and figures make the overall message clearer. Some minor remaining points can be found below.

Reviewer #3: (No Response)

**Part II – Major Issues: Key Experiments Required for Acceptance**

Reviewer #2: (No Response)

Reviewer #3: (No Response)

**Part III – Minor Issues: Editorial and Data Presentation Modifications**

Reviewer #2: 1) In Figure S2B and S3C, include statistics. In Figure S5B and S6C legend, the statistical test performed should be indicated. Indeed, in most of the manuscript, Wilcoxon tests are used, but with 4 donors here, it is not possible to use such a test.

2) Although the new Fig S7 is appreciated, an overlay of 4 histograms is difficult to read. Maybe presenting an offset histogram or plotting the Ki67+ percentages would make this figure clearer.

3) In the abstract: “FTY720 reduces establishment of latent HIV infection” might be confusing given the new Fig S6 that shows that FTY720 does not impair the seeding of the viral reservoir per se. This could be rephrased to avoid any confusion.

4) The description of Fig 2 is sometimes misleading. Indeed, the reduction in integrated HIV DNA at day 17 and the decrease of p24+ cells upon reactivation at day 19 may be the consequence of the lower levels of productive infection observed at day 10 and 13. Concluding that FTY720 leads to “a reduction of the seeding of the latent reservoir in primary CD4 T cells” may be an over statement, given the results of Fig S6. The authors should discuss that the reduction is likely a consequence of a lower productive infection rather than seeding of the reservoir.

5) In the description of Fig S6 in the text, the authors may mention that the results indicate that in this model, latency is established during ART but is not impacted by FTY720.

6) Similarly, in the discussion “Therefore, our results suggest that FTY720 may be useful as a strategy to limit the size of the latent reservoir if used concomitantly with ART initiation”. This hypothesis is not supported by the novel Fig S6.

7) In Fig 3, presenting examples of fusion plots before the graph with all the donors would facilitate reading. Fig 3B and 3C are somewhat redundant.

8) In Fig 5A and 6A, blot images should be surrounded with a black line to make cuts clearer. Plaese assign the molecular weights to the corresponding panel. Molecular weights should be added in Fig 6A and S8.

9) The authors should discuss the differences with the study of Duquenne C et al. (2017) where it was shown that S1PR1 agonists reverse latency, contrary to the results presented here.

Reviewer #3: (No Response)

PLOS authors have the option to publish the peer review history of their article (what does this mean?). If published, this will include your full peer review and any attached files.

Reviewer #2: No

Reviewer #3: No
---

## [Editor Report · Decision Letter 2]

3 Jun 2020

Dear PhD Bosque,

We are pleased to inform you that your manuscript 'Fingolimod inhibits multiple stages of the HIV-1 life cycle' has been provisionally accepted for publication in PLOS Pathogens.

Best regards,

Daniel C. Douek

Associate Editor

PLOS Pathogens

Susan Ross

Section Editor

PLOS Pathogens

Kasturi Haldar

Editor-in-Chief

PLOS Pathogens

orcid.org/0000-0001-5065-158X

Michael Malim

Editor-in-Chief

PLOS Pathogens

orcid.org/0000-0002-7699-2064
---

## [Editor Report · Acceptance letter]

13 Jul 2020

Dear PhD Bosque,

We are delighted to inform you that your manuscript, "Fingolimod inhibits multiple stages of the HIV-1 life cycle," has been formally accepted for publication in PLOS Pathogens.

Best regards,

Kasturi Haldar

Editor-in-Chief

PLOS Pathogens

orcid.org/0000-0001-5065-158X

Michael Malim

Editor-in-Chief

PLOS Pathogens

orcid.org/0000-0002-7699-2064